# 3D NoC Low-Power Mapping Optimization Based on Improved Genetic Algorithm

**DOI:** 10.3390/mi12101217

**Published:** 2021-10-06

**Authors:** Yu Gan, Hong Guo, Ziheng Zhou

**Affiliations:** College of Computer Science and Technology, Wuhan University of Science and Technology, Wuhan 430065, China; rorchach2k@gmail.com (Y.G.); zzh_99_wuhan@foxmail.com (Z.Z.)

**Keywords:** 3D NoC, low power consumption, mapping algorithm, improved genetic algorithm, population optimization, global optimization

## Abstract

Power optimization is an important part of network-on-chip(NoC) design. This paper proposes an improved algorithm based on genetic algorithm on how to properly map IP (Intellectual Property) cores to 3D NoC. First, in view of the randomness of the traditional genetic algorithm in individual selection, an improved greedy algorithm is used in the initial population generation stage to make the generated individuals reach the optimal. Secondly, in view of the weak local optimization ability of the traditional genetic algorithm and prone to premature problems, the simulated annealing algorithm is added in the crossover operation stage to make the offspring reach the global optimum. The experimental results show that compared with the traditional genetic algorithm, the algorithm has better convergence and low power consumption performance, which can quickly search for a better solution, in the case of a large number of cores (124 IP cores), the average power consumption can be reduced by 42.2%.

## 1. Introduction

As the mainstream design technology of next-generation IC, NoC will be the inevitable trend of system-on-chip development [1]. Although NoC breaks the many limitations faced by system-on-chip(SoC), NoC still has inevitable defects, and is still subject to various restrictions from the planar structure. For example, subject to layout conditions, 2D NoC cannot guarantee to shorten the distance of its critical path by making key components adjacent, thereby reducing the transmission delay time. The emergence of 3D NoC precisely solves this limitation [2]. In recent years, the rise of Through Silicon Via (TSV) technology has greatly promoted the development of three-dimensional integrated circuit technology, so 3D NoC has become a hot spot in the industry [3]. With the rapid development of NoC technology and the gradual maturity of 3D technology, more and more research institutions have begun to join the field of 3D NoC, and many listed companies have also invested a lot of manpower and material resources in this field [4]. 3D NoC has attracted widespread attention from academia and industry.

At present, the related research of 3D NoC mainly focuses on the topological structure, quality of service, routing algorithm and communication power consumption of 3D NoC [5,6]. With the increasing number of IP cores integrated on the processor, the power consumption of the chip continues to increase. In 3D NoC, power consumption has a great impact on system performance, so how to reduce power consumption has become a key issue in 3D NoC design.

Many international experts and scholars have proposed many ways to reduce power consumption for 3D NoC in different aspects and levels, which are mainly reflected in logic signal design, routing algorithm optimization and design, and low-power mapping of network topology [7]. 3D NoC mapping is to assign the processing units in the task feature map to the resource nodes of NoC architecture on the basis of the given task feature map and NoC architecture and under certain constraints. The mapping algorithm determines the actual location of the IP core mapped to the 3D NoC architecture. A good mapping algorithm can improve the fault tolerance of 3D NoC and reduce the power consumption and delay of the system. Therefore, it is necessary to study the 3D NoC mapping algorithm from different perspectives [8,9,10].

3D NoC mapping algorithms can be divided into two categories: the traditional mapping optimization algorithm used in the early days and the new intelligent mapping algorithm based on heuristics. Traditional mapping algorithms mainly include exhaustive method, linear programming algorithm, and branching algorithm, which mainly solve the situation where the early structure of the on-chip network is simple and the application is rare [11]. Most of the current research focuses on intelligent algorithms, which are search mechanisms developed to solve optimization problems by imitating the laws of nature or some biological characteristics, such as genetic algorithm, ant colony algorithm, tabu search algorithm, particle swarm algorithm, and simulated annealing algorithm [12]. The results of traditional mapping algorithms are relatively accurate, but as the application scenarios become much more complex, the amount of calculation has been too large. Meanwhile it is even difficult to find the final result based on the existing computing capabilities. Therefore, there is no need to find all the solutions, as long as the intelligent algorithm that meets the required better and feasible solution is quickly recognized by the majority of scholars, it has become a hot spot in the research of on-chip network mapping algorithms in recent years [13]. This work intends to combine traditional intelligent algorithms with other classic algorithms through the study of the 3D NoC mapping model, and propose an optimization algorithm that can avoid the problems of premature convergence and local optimization of traditional intelligent algorithms, and can effectively reduce the power consumption of NoC systems.

## 2. 3D NoC Mapping Model

### 2.1. 3D NoC Mesh Architecture

The 3D NoC mapping platform is composed of resource nodes, routing, and physical links. This article uses a regular 3D NoC topology, as shown in Figure 1. The size of each layer of the 3D NoC is the same, and the routing in the vertical direction is connected through TSV (through silicon vias) [14,15]. The size of the 3D NoC mapping platform is N × N × L, N × N represents the number of resource nodes of each layer of the mapping platform, L represents the number of layers. To sum up, N × N × L is the number of resource nodes that can be mapped by the 3D NoC. The size of the 3D NoC architecture in Figure 1 is 3 × 3 × 3, and there are 27 resource nodes that can be mapped. According to the number of connected links, resource nodes can be divided into four types, with the number of connected links being 6, 5, 4, and 3 respectively. For example, resource node No. 14 has 6 links, and resource node No. 1 has 3 links. These data links connect adjacent resource nodes on a horizontal or vertical level to ensure information exchange between resource nodes.

### 2.2. 3D NoC Mapping

Network mapping on a chip refers to assigning the logical IP cores with known communication relations and traffic volumes on a given task characteristic graph (TCG) to the corresponding resource nodes of the architecture characteristic graph (ARCG) under specified system constraints, so that various applications can be completed smoothly and efficiently under the constraints [16,17]. The mapping process is shown in Figure 2.

**Definition** **1.**
*The task graph TCG(V, T) is an acyclic directed graph, the vertex $v_{i}\in V$ represents a task, the directed arc $t_{i,j}\in T$ represents the communication between tasks $t_{i}$ and $t_{j}$, and the weight $\omega_{i,j}$ of $t_{i,j}$ represents the amount of communication data between $t_{i}$ and $t_{j}$. If $t_{i}$ and $t_{j}$ have no communication relationship, then $\omega_{i,j}=0$. The edge set W represents the communication relationship between tasks, and the communication relationship matrix between all tasks is:*

(1)
$$W=\left[\omega_{i,j}\right]=\left[\begin{array}{ccc} 0 & \cdots & \omega_{0,n-1}\\ \vdots & \ddots & \vdots \\ \omega_{n-1,0} & \cdots & 0 \end{array}\right]$$



Among them, 0≤i≤n−1, 0≤j≤n−1.

**Definition** **2.**
*Structural feature graph ARCG(E, P) is also a directed graph, vertex $e_{i}\in E$ represents a resource node, edge $p_{i,j}\in P$ represents the communication path between $p_{i}$ and $p_{j}$, and the weight $f_{i,j}$ of $p_{i,j}$ represents the amount of communication data between them. If $p_{i}$ and $p_{j}$ have no communication relationship, then $f_{i,j}=0$.*


**Definition** **3.**
*When the task graph TCG(V, T) and the structural feature graph ARCG(E, P) are the same in size, that is, V=E, it meets the requirements of one-to-one mapping.*


**Definition** **4.**
*For 3D-Mesh NoC, $M_{d_{i,j}}$ represents the jump distance (Manhattan distance) from $e_{i}$ to $e_{j}$, namely:*

(2)
$$M_{d_{i,j}}=\left|x_{i}-x_{j}\right|+\left|y_{i}-y_{j}\right|+\left|z_{i}-z_{j}\right|$$



Therefore, the communication volume $f_{i,j}$ between resource nodes $e_{i}$ to $e_{j}$ can be expressed as:(3)$$f_{i,j}=\omega_{i,j}\times M_{d_{i,j}}$$

Therefore, for a given task graph TCG(V, T) and feature structure graph ARCG(E, P), when it satisfies Definition 3, the objective function of the NoC mapping problem can be defined as:(4)$$\{\begin{array}{c} map\left(V\to E\right)\;\;\;\;\;\;\;\;\;\;\;\;i\in \left|V\right|,\;j\in \left|V\right|\\ s.t.\;\;m\left(e_{i}\right)=v_{i}\;\;\;\;\;\;\;\;\;\forall e_{i}\in E,\;\exists v_{i}\in V \end{array}$$

### 2.3. 3D NoC Power Consumption Model

The power consumption of a 3D NoC system is mainly the power consumption generated by data communication between arbitrary resource nodes. Therefore, in order to maximize the reduction of power consumption, this paper adopts the power consumption model proposed in [18], which is, the energy consumption of a basic length of 1-bit data transmission on the NoC can be expressed as:(5)$$E_{bit}=E_{S_{bit}}+E_{B_{bit}}+E_{W_{bit}}+E_{L_{bit}}$$

Among them, $E_{S_{bit}}$, $E_{B_{bit}}$, $E_{W_{bit}}$, and $E_{L_{bit}}$ respectively represent the energy consumed on the crossbar switch, buffer, internal interconnection line, and adjacent routing node links. Since $E_{B_{bit}}$ and $E_{W_{bit}}$ are very small in actual situations, they can usually be ignored. Because the platforms used in this article are homogeneous, the power consumption generated by each router with the same amount of data is approximately equal, so the Formula (5) is changed to:(6)$$E_{bit}=E_{S_{bit}}+E_{L_{bit}}$$

Therefore, the formula for calculating the energy consumed by 1-bit data from node $e_{i}$ to node $e_{j}$ is:(7)$$E_{bit}^{n_{i,j}}=\left(n_{i,j}+1\right)\times E_{S_{bit}}+n_{i,j}\times E_{L_{bit}}$$

Among them, $n_{i,j}$ represents the number of nodes passed, that is, the number of routers traversed on the transmission path from node $e_{i}$ to node $e_{j}$, generally measured by Manhattan distance. On the premise of adopting the shortest path routing, $n_{i,j}=M_{d_{i,j}}$. That is, the Formula (7) can be expressed as:(8)$$E_{bit}^{n_{i,j}}=\left(M_{d_{i,j}}+1\right)\times E_{S_{bit}}+M_{d_{i,j}}\times E_{L_{bit}}$$

Then the communication power consumption of the entire NoC system is:(9)$$E=\overset{j\leq \left|V\right|}{\underset{i\leq \left|V\right|}{\sum }}\left[\left(M_{d_{i,j}}+1\right)\times E_{S_{bit}}+M_{d_{i,j}}\times E_{L_{bit}}\right]\times \omega_{i,j},\;\;\;i\in \left|V\right|,\;j\in \left|V\right|$$

From Equation (3), we can get:(10)$$E=\overset{j\leq \left|V\right|}{\underset{i\leq \left|V\right|}{\sum }}f_{i,j}\times \left(E_{S_{bit}}+E_{L_{bit}}\right)+\overset{j\leq \left|V\right|}{\underset{i\leq \left|V\right|}{\sum }}\omega_{i,j}\times E_{S_{bit}},\;\;\;i\in \left|V\right|,\;j\in \left|V\right|$$

Among them, all are constants except $\overset{j\leq \left|V\right|}{\underset{i\leq \left|V\right|}{\sum }}f_{i,j}$, so the objective function of low-power mapping can be expressed as:(11)$$\{\begin{array}{c} \min\sum_{i\leq \left|V\right|}^{j\leq \left|V\right|}f_{i,j}\;\;\;\;\;\;\;\;\;\;\;i\in \left|V\right|,\;j\in \left|V\right|\\ s.t.\;\;m\left(e_{i}\right)=v_{i}\;\;\;\;\;\;\;\;\;\forall e_{i}\in E,\;\exists v_{i}\in V \end{array}$$

Combining the above formula, it can be seen that the key factor of low-power mapping is $f_{i,j}$, so we can consider reducing power consumption by reducing $f_{i,j}$ in the NoC mapping process. This objective function with power optimization as the goal will be the basis for the realization of the mapping algorithm proposed later.

Due to the emergence of TSV technology, the line length of 3D NoC in the vertical direction is much smaller than the length in the horizontal direction. Therefore, to transmit the same amount of data, the energy consumption in the vertical direction is much smaller than the energy consumption in the horizontal direction.

## 3. 3D NoC Low-Power Mapping Based on Improved Genetic Algorithm

### 3.1. Proposal of Improved Genetic Algorithm

The genetic algorithm is to construct an adaptive value function according to the solution of the problem to generate a population composed of multiple solutions. Individuals in the population are selected for genetic operation to generate a new population according to the adaptive value. When the evolution reaches the termination condition, individuals with good adaptive value are selected as the solution of the problem. The idea of genetic algorithm is easy to implement, the algorithm is highly efficient, has strong global search ability, and is suitable for solving discrete problems. Therefore, genetic algorithm is very suitable for solving 3D NoC mapping problems [19].

The initial population of the traditional genetic algorithm is generated randomly, the individual distribution of the initial population is not uniform, and the quality of the population is low. At the same time, the traditional genetic algorithm has strong global search ability, but poor local search ability, and easy to fall into the trap of precocity, it takes a long time to get the optimal solution. Therefore, the traditional genetic algorithm needs to be improved to solve the 3D NoC mapping problem [20,21,22].

In view of the shortcomings of traditional genetic algorithm that generate the initial population with large randomness and low population quality, this paper adds an improved greedy algorithm in the selection operation stage; in view of the shortcomings of traditional genetic algorithm of strong global optimization ability and poor local optimization ability, this work adds a simulated annealing algorithm in the crossover operation stage. The improved genetic algorithm realizes the optimization of the initial population and the global optimization of the offspring.

The process of improving genetic algorithm is shown in Figure 3.

Implementation steps of improved genetic algorithm:

Step l. Initialize the population with an improved greedy algorithm.

Step 2. When *i* is less than or equal to the maximum number of iterations, continue to step 3, otherwise it ends, and the optimal solution is returned.

Step 3. Calculate the individual fitness value.

Step 4. The male parent is selected according to the fitness value, the male parent with the higher fitness value is selected, and the male parent with the lower fitness value is eliminated.

Step 5. The simulated annealing algorithm is used to crossover the parent chromosomes to generate offspring.

Step 6. Perform mutation operations on the offspring’s chromosomes to generate new offspring.

Step 7. Obtain the optimal individual, update the population, and return to step 2 until the optimal solution is produced.

### 3.2. Population Individual Vector Representation

Individual vector representation is the first problem to be solved when designing genetic algorithms, and it is also a key step in it. The way of vector representation will affect the calculation methods of crossover and mutation operators, and determines the efficiency of genetic evolution.

Since each processing unit can send data to any other processing unit, the IP core in the TCG can be mapped to any processing unit. In the 3D NoC mapping, the population individuals are coded by integers, assuming that the individual *X* = (*x*_1_, *x*_2_,…, *x*_*n*_), *n* is the number of IP cores of the TCG, and *x_i_* is the number of the IP core. The code of individual *X* is a sequence from 1 to *n*, and a mapping scheme can be obtained by decoding the code of individual *X* [23].

A simple task graph PIP is shown in Figure 4. Take this graph as an example to illustrate the vector representation of individuals.

The IP cores in the task map are represented by numbers 0 to 7, mapping PIP to a 2 × 2 × 2 3D NoC. Each processing unit on the 3D NoC is represented by numbers *t*_0_ to *t*_7_, *t*_0_ to *t*_3_ represent the first layer, and *t*_4_ to *t*_7_ represent the second layer, as shown in Figure 5.

Individual *X* = (2, 3, 4, 1, 0, 5, 7, 6) means to map IP cores (2, 3, 4, 1, 0, 5, 7, 6) to processing units (0, 1, 2, 3, 4, 5, 6, 7), the mapping result is shown in Figure 6.

After all IP cores are mapped to the 3D NOC, calculate the total amount of communication between all nodes on the 3D NoC. The genetic algorithm calculates its fitness value based on the total amount of communication on the 3D NoC. The larger the total amount of communication, the smaller the fitness value. The genetic algorithm performs genetic operations based on the fitness value.

### 3.3. Initial Population Optimization Based on Improved Greedy Algorithm

The initial population selection of traditional genetic algorithms is generally random, the quality of the obtained population is usually low while the iteration speed is slow. In response to this situation, this paper proposes an initialization method based on an improved greedy algorithm to generate population Pop, which can greatly increase the iteration speed of the population, effectively reduce the amount of data communication, and reduce power consumption [24].

The initial population optimization process of the improved greedy algorithm is shown in Figure 7.

The initial population optimization steps of the improved greedy algorithm:

Step l. Randomly generate a number *i* (*i* = 1, 2, …, *n*), and then map this random number *i* to the first position of individual *X*, and use an improved greedy algorithm to generate an array of the smallest spanning tree with *i* as the first node.

Step 2. Initialize the available set *P* = {1, 2, …, *n*}, where *n* is the number of IP cores of the TCG, and delete *i* from the available set *P*.

Step 3. Traverse the number *n* in the available set *P*, put *n* into all available positions of the individual *X*, and calculate the fitness value of *X* after putting them in these positions. Find the maximum fitness value *f_it_* from these fitness values, mark the position *m* of the maximum fitness value *f_it_* after *n* is placed in *X*, and save the element <*n*, *m*, *f_it_*> to the set *F*.

Step 4. Traverse the set *F*, find a group of elements <*n*, *m*, *f_it_*> with the largest *f_it_* value, put *n* into the *m*-th position of the individual *X*, delete *n* from the available set *P*, and delete *m* from the available positions.

Step 5. Repeat Step 3 and Step 4 until the available set *P* is empty. When *P* is empty, it means that a new individual *X* has been produced, and *X* is added to the temporary population Temppop.

Step 6. Repeat Step 5 above to generate *n* individuals.

Step 7. Exchange any two coordinate numbers of individual *X* to generate a neighbor individual of *X*. Use this method to generate 20 neighbor individuals for the individual *X* generated by the above steps, and put these neighbor individuals into the temporary population Temppop.

Step 8. Repeat all the above steps *n* times to generate *n* individuals obtained by the improved greedy algorithm and 20 neighbor individuals of these individuals.

Step 9. Select multiple individuals with the largest fitness value from Temppop and put them into Pop as the initial population.

### 3.4. Population Global Optimization Based on Simulated Annealing Algorithm

The simulated annealing algorithm is based on the Monte-Carlo iterative solution strategy to find the optimal solution randomly. It can probabilistically jump out of the local optimal solution during the solution process and obtain the final global optimal solution. The simulated annealing algorithm overcomes the poor local optimization ability of the traditional genetic algorithm and its premature phenomenon. Combining it with the traditional genetic algorithm will better exert the global optimization ability of the traditional genetic algorithm and the local optimization ability of the simulated annealing algorithm [25].

Based on this consideration, this work adds the simulated annealing algorithm to the crossover operation stage of the traditional genetic algorithm to ensure that the generated offspring will not fall into the local optimum, so that the population jumps out of the local optimum and finally reaches the global optimum.

The population global optimization process of the simulated annealing algorithm is shown in Figure 8.

Suppose the fitness value of the newly generated individual is *f*, and the threshold of change is *f*^′^. When *f* > *f*^′^, the new individual is accepted; otherwise, the new individual is accepted with a certain probability $P=exp\left(\left(f-f^{\prime }\right)/T\right)$. Among them, *T* represents temperature.

The population global optimization steps of the simulated annealing algorithm:

Step 1. Set the starting temperature *T*_0_ and the lowest temperature *T*_final_.

Step 2. Initialize temperature *T* = *T*_0_, *i* = 0.

Step 3. When *T* > *T*_final_, execute the next step; otherwise, it ends and returns the optimal solution.

Step 4. If *i* ≤ the number of crossover operations, execute the next step; otherwise, it ends and returns to the optimal solution.

Step 5. Select *n* pairs of individuals from the population as male parents, and perform the following operations on each male parent:

Step 5.1. The crossover operation is performed on the parents *P*_1_ and *P*_2_ to generate offspring *S*_1_ and *S*_2_, and the fitness values of *S*_1_ and *S*_2_ are calculated.

Step 5.2. If $f_{s_{1}}>f_{p_{1}}$, $f_{s_{2}}>f_{p_{2}}$, replace *P*_1_ and *P*_2_ with *S*_1_ and *S*_2_; otherwise, keep *P*_1_ and *P*_2_ with the probability of $P=exp\left(\left(f_{s_{1}}-f_{s_{2}}\right)/T\right)$.

Step 6. Cool down according to method $T=T_{0}/lg\left(1+i\right)$, *i* = *i* + 1.

Step 7. Go to Step 4.

Among them, *T*_0_ and *T*_final_ represent the initial temperature and the end temperature, respectively.

## 4. Simulation Experiment and Result Analysis

### 4.1. Simulation Platform and Parameter Design

#### 4.1.1. Simulation Platform Selection

The simulation experiment is under the Ubuntu 14.04 operating system, using C++ language to write the algorithm implementation program; under the Codeblocks 13.11 environment, using Access Noxim 0.2 as the simulation software to simulate the 3D NoC mapping algorithm [26].

#### 4.1.2. Topology and Routing

(1)Topology Selection

The 3D Mesh structure is obtained by directly extending the 2D Mesh to the three-dimensional structure, so its structure is relatively simple. At the same time, it also has certain advantages in terms of layout and routing. As the TSV technology is used in the vertical direction, the overall wiring length is reduced and the transmission efficiency is improved. Therefore, this work chooses to use the 3D Mesh structure for experiments [15].

(2)Routing

In terms of routing algorithm, this article adopts XYZ dimension order routing algorithm: XYZ routing algorithm is simple to implement and it is the most commonly used routing algorithm in 3D NoC [27].

#### 4.1.3. Parameter Setting

(1)Algorithm Parameter Settings

Suppose the experimental population size is 200, the number of genetic iterations is 100, the crossover rate is 0.9, and the mutation rate is 0.02.

(2)Parameter Setting of Simulation Software

The data packet is injected using Memory-less Poisson Distribution, and the packet injection rate is 0.02; the size of the data packet ranges from 2 flits to 10 flits; the buffer size of each channel of the router is 8 flits. The simulation software counts the total power consumption of 5000 cycles.

#### 4.1.4. Hardware Operating Environment

The hardware environment configuration of the simulation experiment: A PC with an Intel Core i5-3470 CPU, 3.2 GHz main frequency, and 8 GB memory is used.

### 4.2. Simulation Experiment Comparison and Analysis

In the experiment, two mapping algorithms, improved genetic algorithm and traditional genetic algorithm, were used to compare and analyze the convergence speed and power consumption of a given task graph and its mapping model.

#### 4.2.1. Experimental Task Graph and Its Mapping Model

The simulation experiment uses three classic task maps MWD (Multi Window Displayer), VOPD (Video Object Plane Decoder), and DVOPD (Double Video Object Plane Decoder) for simulation experiments. Among them, MWD has 12 nodes, which are mapped to a 2 × 2 × 3 3D NoC. VOPD has 16 nodes, which are mapped to a 2 × 2 × 4 3D NoC. DVOPD has 32 nodes, which are mapped to a 2 × 2 × 4 3DNoC. As shown in Figure 9 [28].

#### 4.2.2. Convergence Speed Comparison Based on Classic Task Graph

Three classic task graphs MWD, VOPD and DVOPD are simulated, which MWD has 12 nodes, VOPD has 16 nodes, and DVOPD has 32 nodes. The number of simulation iterations is 100. The abscissa in the figure represents the number of iterations, and the ordinate represents the fitness value. The fitness value is negatively correlated with the communication volume of NoC. The lower the communication volume, the greater the fitness value.

(1)Convergence Rate Analysis for MWD

The convergence speed comparison of MWD iteration 100 times is shown in Figure 10. It can be seen from Figure 10 that the fitness value of the improved genetic algorithm at the beginning of the iteration is slightly greater than that of the basic genetic algorithm, and the fitness value of the traditional genetic algorithm changes very little during the evolution process. This shows that when the task graph is small, the traditional genetic algorithm is easy to fall into the local optimum.

(2)Convergence Rate Analysis for VOPD

The convergence speed comparison of VOPD iteration 100 times is shown in Figure 11. It can be seen from Figure 11 that the fitness value of the improved genetic algorithm at the beginning of the iteration is much greater than that of the traditional genetic algorithm. At the same time, because the improved genetic algorithm improves the selection method of the initial population, the improved genetic algorithm can obtain the optimal solution in a few iterations, and the convergence speed is significantly faster than that of the traditional genetic algorithm.

(3)Convergence Speed Analysis for DVOPD

The comparison of the convergence speed of DVOPD iteration 100 times is shown in Figure 12. It can be seen from Figure 12 that since the improved genetic algorithm has a higher quality at the time of the initial solution, the solution obtained by the improved genetic algorithm from the iteration is much better than that of the traditional genetic algorithm. In addition, the simulated annealing algorithm enhances the local optimization ability of the offspring population, and can quickly generate an optimized offspring population. Therefore, the improved genetic algorithm can obtain the optimal solution in a smaller number of iterations, and the convergence speed is significantly faster than that of the traditional genetic algorithm.

Comparative experiments show that as the number of iterations increases, the improved genetic algorithm has a greater advantage in convergence speed. With the increase in the number of IP cores and the amount of communication, this advantage will become more obvious.

#### 4.2.3. Power Consumption Comparison Based on Classic Task Graph

Because the traditional genetic algorithm has a certain degree of randomness, in the experiment, the traditional genetic algorithm and the improved genetic algorithm are used to solve the three task graphs 10 times respectively, and the average value is taken.

(1)Power Consumption Analysis for MWD

The experimental results of the two algorithms of MWD are shown in Figure 13. It can be seen from Figure 13 that compared with the traditional genetic algorithm, the average power consumption of the improved genetic algorithm is reduced by 1.52%, the maximum power consumption is reduced by 6.45%, and the minimum power consumption is basically the same. Although the power consumption of the improved genetic algorithm has been reduced, the decrement is small. This is due to the fact that when the number of nodes is small, the two algorithms can quickly find a better solution under a certain number of iterations.

(2)Power Consumption Analysis for VOPD

The experimental results of the two VOPD algorithms are shown in Figure 14. It can be seen from Figure 14 that compared with the traditional genetic algorithm, the average power consumption of the improved genetic algorithm is reduced by 8.35%, the maximum power consumption is reduced by 16.74%, and the minimum power consumption is reduced by 5.02%. In terms of maximum power consumption, the power consumption of the improved genetic algorithm has been significantly reduced. This is because traditional genetic algorithm populations are randomly generated populations whose quality is relatively low. After adopting improved genetic algorithms, the quality of the initial populations is significantly improved compared to the randomly generated initial populations.

(3)Power Consumption Analysis for DVOPD

The experimental results of the two DVOPD algorithms are shown in Figure 15. It can be seen from Figure 15 that compared with the traditional genetic algorithm, the average power consumption of the improved genetic algorithm is reduced by 23.58%, the maximum power consumption is reduced by 26.68%, and the minimum power consumption is reduced by 25.17%. With the increase in the number of IP cores, the power consumption of the improved genetic algorithm has been greatly reduced in all aspects. By improving the quality of the initial population, the convergence speed is accelerated. At the same time, due to the addition of the simulated annealing algorithm in the crossover operation stage, the offspring population is prevented from falling into local optimization. The combined effect of the two makes the improved genetic algorithm have a better low power consumption advantage than the traditional genetic algorithm.

Comparative experiments show that as the number of IP cores increases, the total amount of communication decreases more and more, which means that the system power consumption will decrease more and more. Therefore, the improved genetic algorithm proposed is effective to reduce the power consumption for classic task graphs.

#### 4.2.4. Power Consumption Comparison Based on Random Task Graph

The task generator TGFF is used to generate random task graphs with IP cores of 45, 60, 80, 98, and 124 [29]. For task graphs with different IP core numbers, when the population size is 200, two algorithms are used to solve 10 times, and the average value is taken. The experimental comparison results of power consumption between the improved genetic algorithm and the traditional genetic algorithm are shown in Table 1 and Table 2.

(1)Analysis of Average Power Consumption

When the number of IP cores is 45, the power consumption of the improved genetic algorithm is reduced by 36.8% compared with the traditional genetic algorithm. When the number of IP cores is 60, the power consumption of the improved genetic algorithm is reduced by 39.0% compared with the traditional genetic algorithm. When the number of IP cores is 98, the power consumption of the improved genetic algorithm is reduced by 39.3% compared with the traditional genetic algorithm. When the number of IP cores is 124, the power consumption of the improved genetic algorithm is reduced by 42.2% compared with the traditional genetic algorithm. The average power consumption comparison of the two algorithms is shown in Figure 16.

(2)Analysis of Maximum Power Consumption

When the number of IP cores is 45, the power consumption of the improved genetic algorithm is reduced by 16.3% compared with the traditional genetic algorithm. When the number of IP cores is 60, the power consumption of the improved genetic algorithm is reduced by 37.7% compared with the traditional genetic algorithm. When the number of IP cores is 98, the power consumption of the improved genetic algorithm is reduced by 26.4% compared with the traditional genetic algorithm. When the number of IP cores is 124, the power consumption of the improved genetic algorithm is reduced by 25.6% compared with the traditional genetic algorithm. The maximum power consumption comparison of the two algorithms is shown in Figure 17.

(3)Analysis of Minimum Power Consumption

Compared to the traditional genetic algorithm, when the number of IP cores is 45, the power consumption of the improved genetic algorithm is reduced by 43.6%; when the number of IP cores is 60, the power consumption of the improved genetic algorithm is reduced by 47.8%; when the number of IP cores is 98, the power consumption of the improved genetic algorithm is reduced by 21.0%; and finally when the number of IP cores is 124, the power consumption of the improved genetic algorithm is reduced by 40.5%. The minimum power consumption comparison of the two algorithms is shown in Figure 18.

It can be seen from the above experimental data analysis that when the number of IP cores is small, the reduction in power consumption of the improved genetic algorithm compared with the traditional genetic algorithm is not obvious. This is because when the number of tasks is small, the initial population improvement of the improved genetic algorithm and the selection of good genes are not outstanding, and the algorithm may converge earlier. However, with the increase in the number of IP cores, from the overall trend, the reduction in power consumption of the improved genetic algorithm is gradually greater than that of the traditional genetic algorithm.

### 4.3. Experimental Analysis Conclusions

Through the above experiments, the following conclusions can be drawn:

In terms of convergence speed, when the task graph is small, the traditional genetic algorithm is easy to fall into local optimization, and the advantages of the improved genetic algorithm over the traditional genetic algorithm are not obvious. As the scale of the task graph increases and the number of iterations increases, the convergence speed of the improved genetic algorithm is faster than that of the traditional genetic algorithm. The maximum fitness value obtained by the improved genetic algorithm in the early stage of the algorithm is larger than that of the traditional genetic algorithm, while the improved genetic algorithm in the later period of the algorithm has a faster convergence speed and a larger fitness value.

In terms of the power consumption of the classic task graph, when the task graph is small, the power consumption of the improved genetic algorithm is sometimes higher than that based on the traditional genetic algorithm. This is because when there are too few nodes in the task graph, the initial population obtained by improving the greedy strategy may sometimes fall into a local optimum. After the task scale gradually increases, this situation will be improved, the advantages of the improved genetic algorithm will be reflected, and the system power consumption will be reduced more significantly.

In terms of power consumption of random task graphs, when the task graph is small, the advantages of improved genetic algorithms over traditional genetic algorithms are not obvious, and the reduction in power consumption is small. This is because when the number of IP cores is small, it is easy to cause the two algorithms to fall into the local optimum. However, judging from the overall trend, as the number of IP cores increases, the advantages of improved genetic algorithms are gradually revealed, and the reduction in power consumption has increased significantly.

In view of the randomness of the traditional genetic algorithm when generating the initial population, adding an improved greedy algorithm in the population initialization stage can optimize the individual of the initial population and improve the quality of the initial population. Then in view of the lack of local optimization ability in the traditional genetic algorithm in the crossover operation stage, adding the simulated annealing algorithm to the crossover operator can realize the global optimization of the offspring population and improve the quality of the offspring population. The improved genetic algorithm proposed not only retains the fast random search characteristics of the traditional genetic algorithm, but also solves the shortcomings of the traditional genetic algorithm in the 3D NoC mapping process. Experimental results show that the improved genetic algorithm makes the layout of IP cores more reasonable, reduces the long-distance communication between IP cores, and significantly reduces the power consumption of the chip.

## 5. Conclusions

This paper first introduces the 3D NoC power mapping algorithm based on traditional genetic algorithm, analyzes the problems of traditional genetic algorithm, and proposes an improved genetic mapping algorithm based on initial population selection and genetic operator selection. After that, for the 3D NoC model, its coding scheme, topology, routing algorithm and genetic parameter settings are given, and the convergence speed and power consumption of the two algorithms are compared for the classic task graph and the random task graph. The experimental results show that compared with the traditional genetic algorithm, the improved genetic algorithm proposed in this paper has a good effect in solving 3D NoC low-power mapping. With the increase in the number of IP cores, improved genetic algorithms have more significant advantages in reducing power consumption, and can better reduce the power consumption of 3D NoC compared to traditional genetic algorithms.

## Figures and Tables

**Figure 1 micromachines-12-01217-f001:**
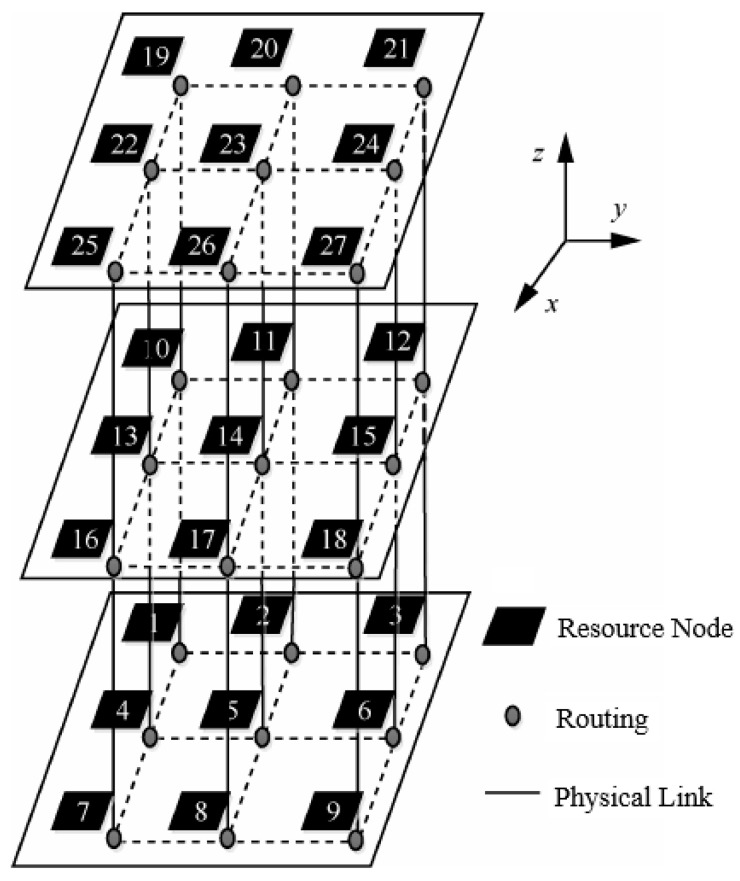
Architecture Characteristic Graph.

**Figure 2 micromachines-12-01217-f002:**
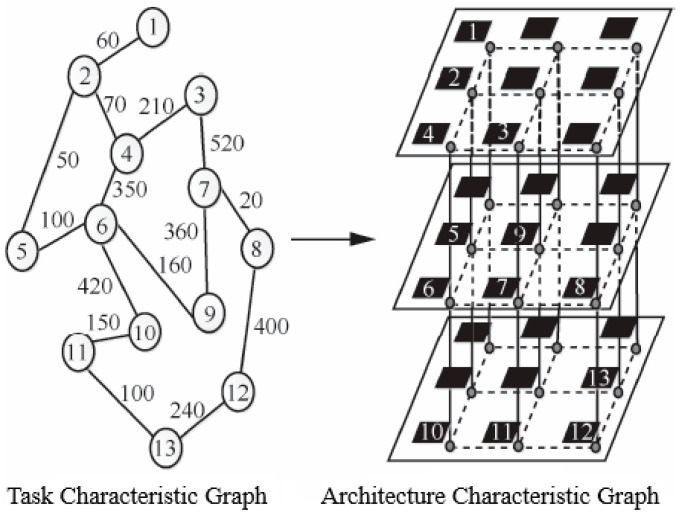
3D NoC mapping.

**Figure 3 micromachines-12-01217-f003:**
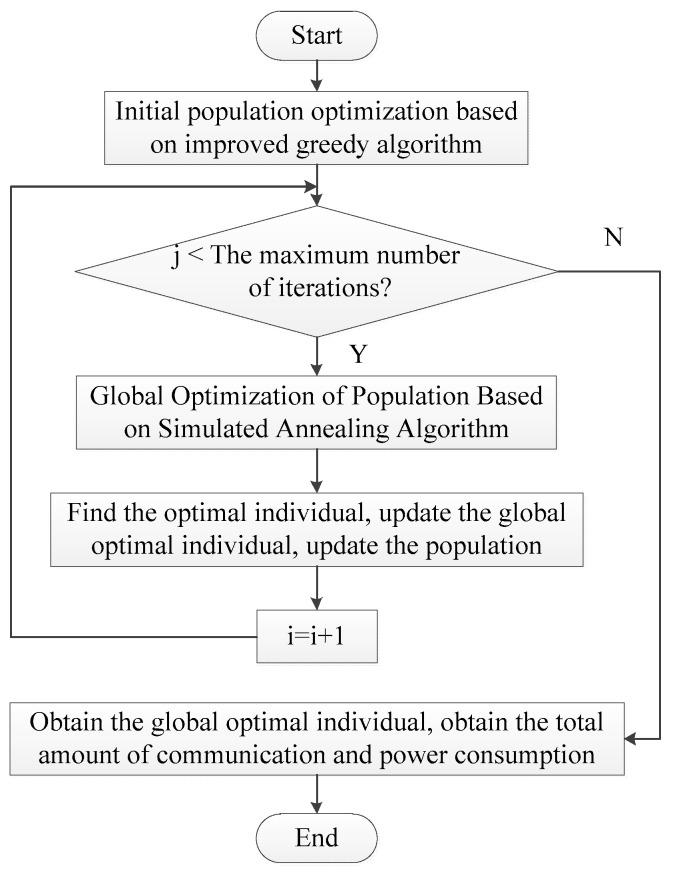
Flow chart of improved genetic algorithm.

**Figure 4 micromachines-12-01217-f004:**
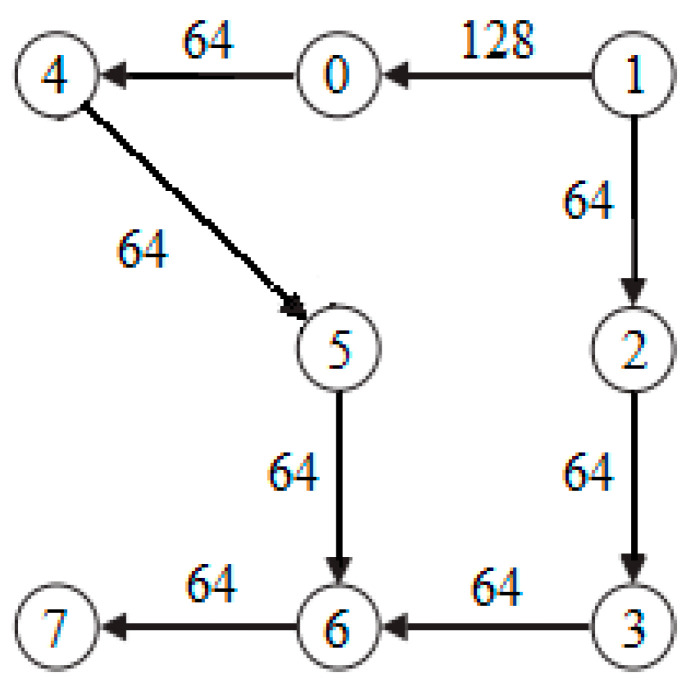
Simple task graph PIP.

**Figure 5 micromachines-12-01217-f005:**
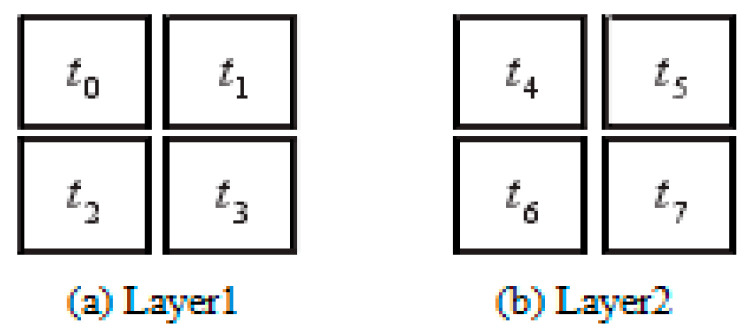
3D NoC topological structure.

**Figure 6 micromachines-12-01217-f006:**
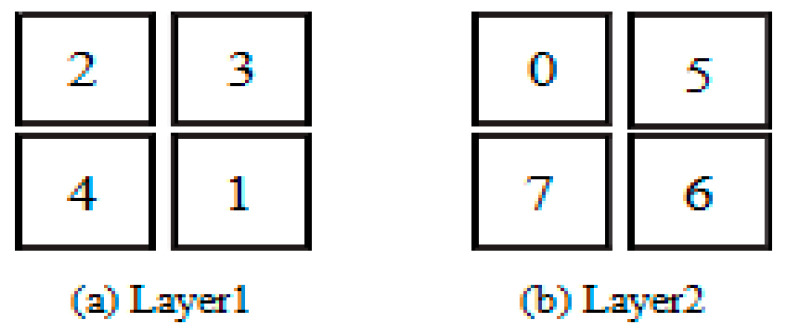
Mapping results.

**Figure 7 micromachines-12-01217-f007:**
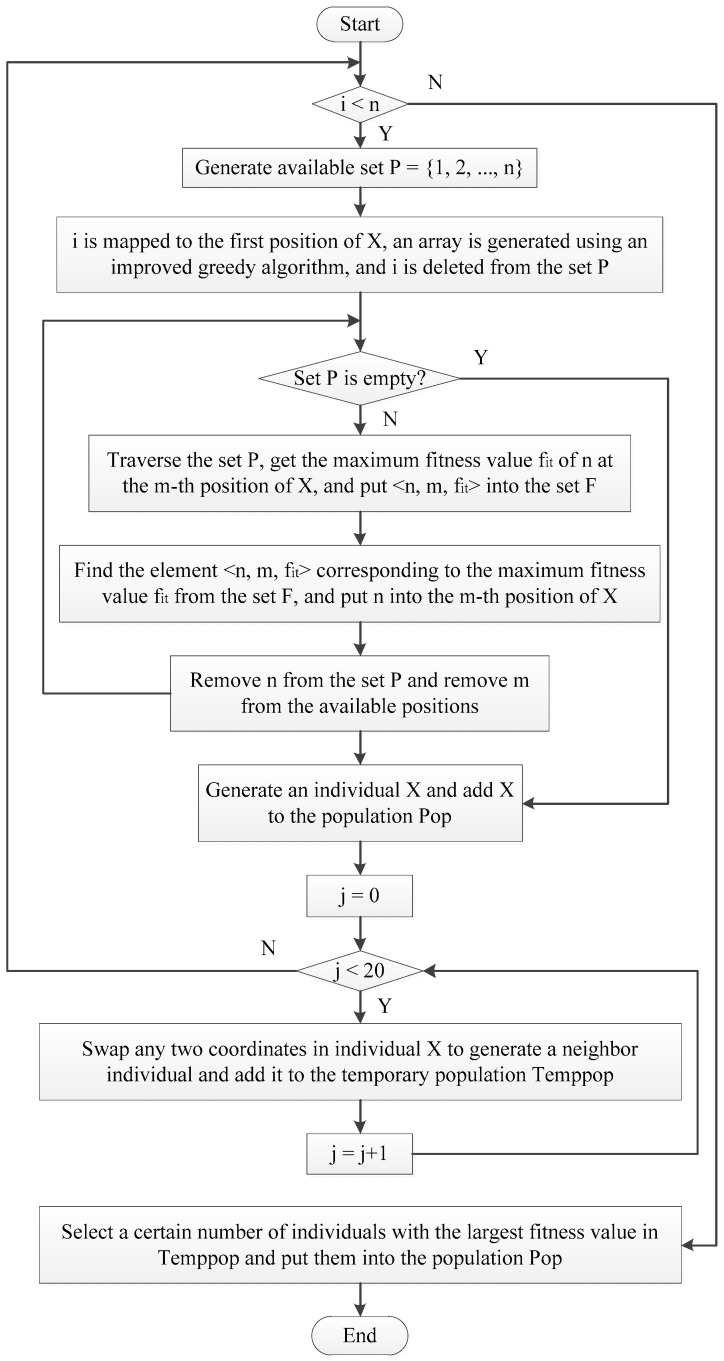
Improved greedy algorithm flow chart.

**Figure 8 micromachines-12-01217-f008:**
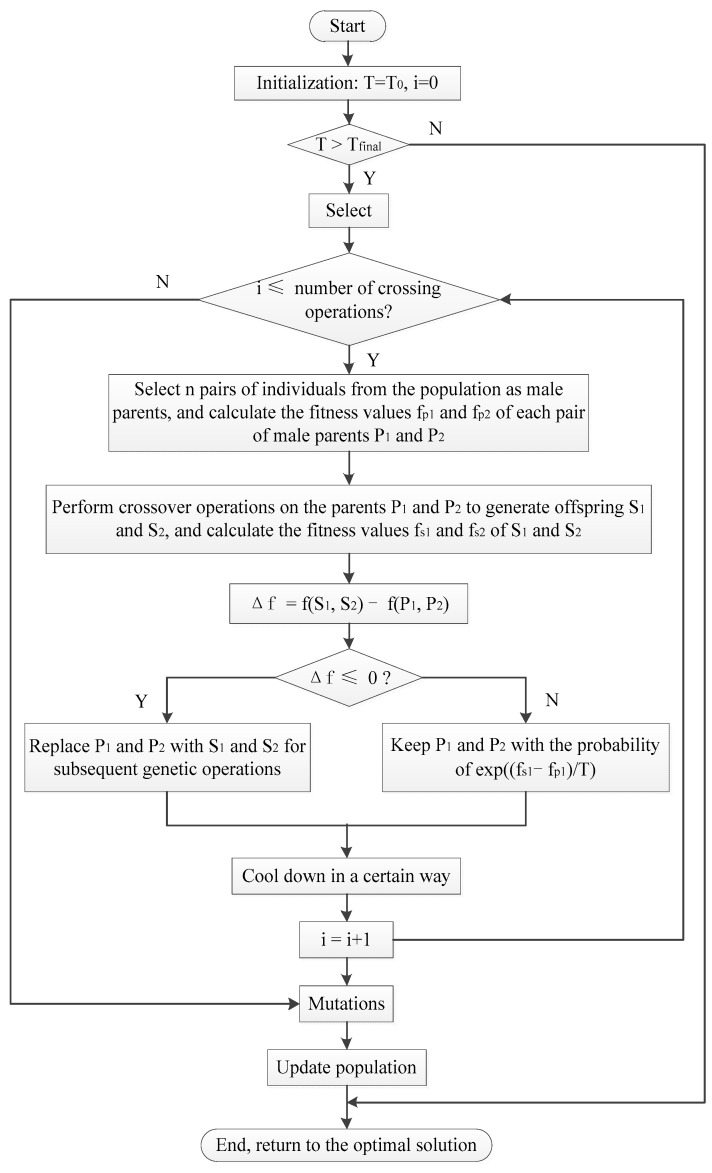
Simulated annealing algorithm flow chart.

**Figure 9 micromachines-12-01217-f009:**
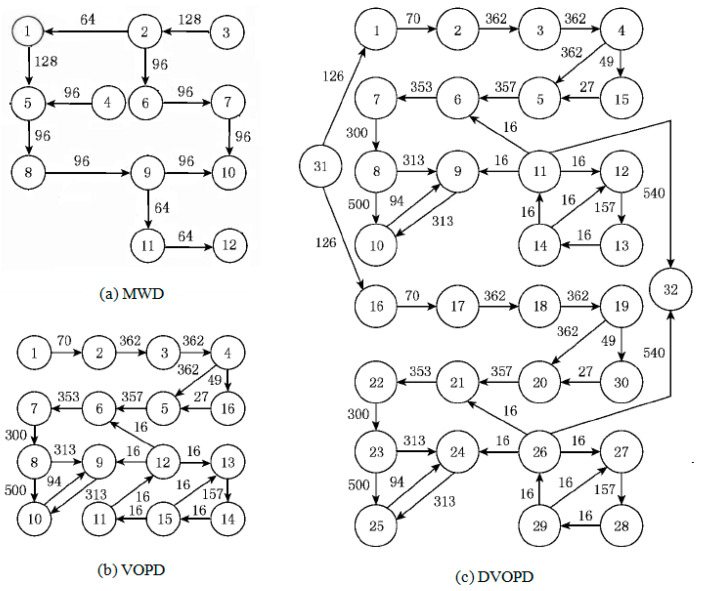
Classic task graph.

**Figure 10 micromachines-12-01217-f010:**
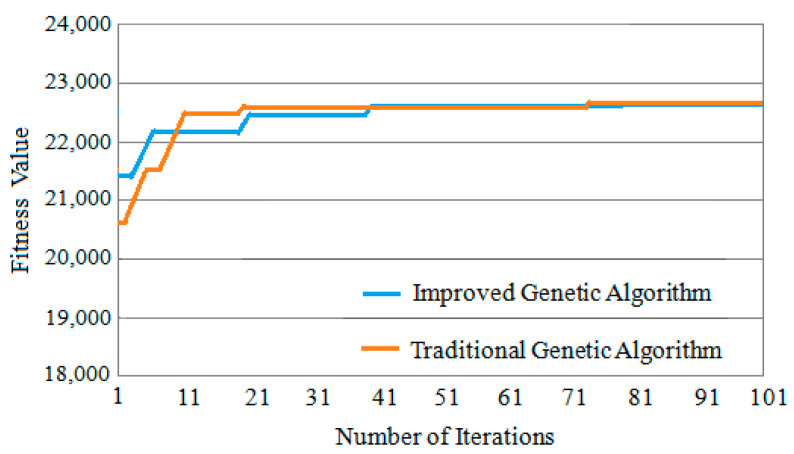
Convergence speed comparison of MWD.

**Figure 11 micromachines-12-01217-f011:**
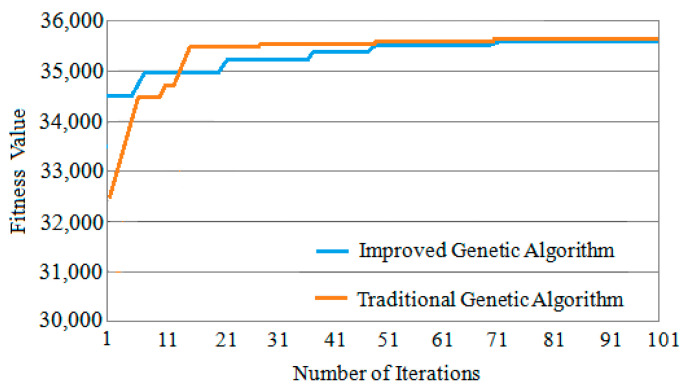
Convergence speed comparison of VOPD.

**Figure 12 micromachines-12-01217-f012:**
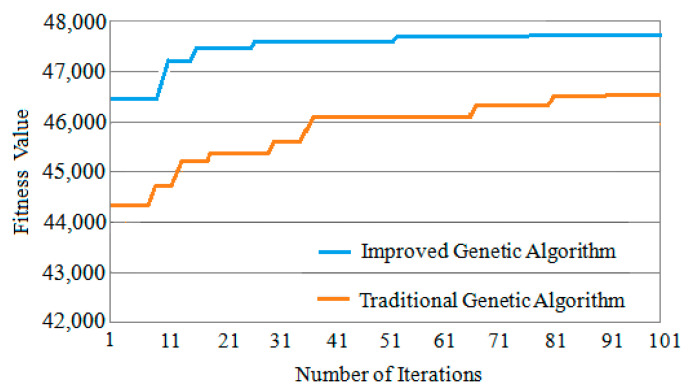
Convergence speed comparison of DVOPD.

**Figure 13 micromachines-12-01217-f013:**
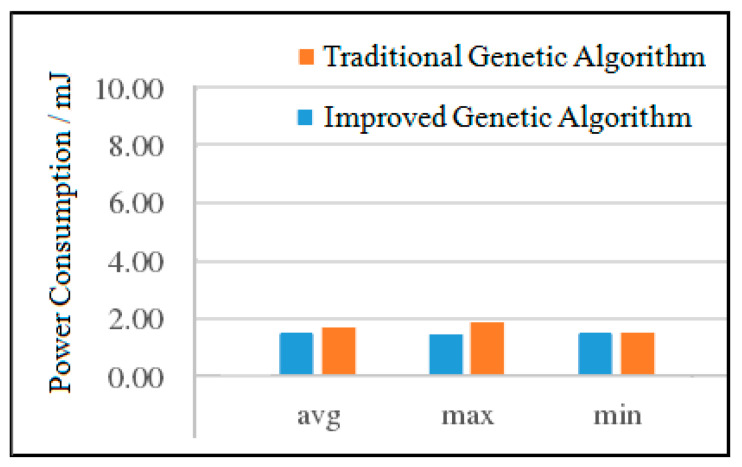
Comparison of the power consumption of the two algorithms for VOPD.

**Figure 14 micromachines-12-01217-f014:**
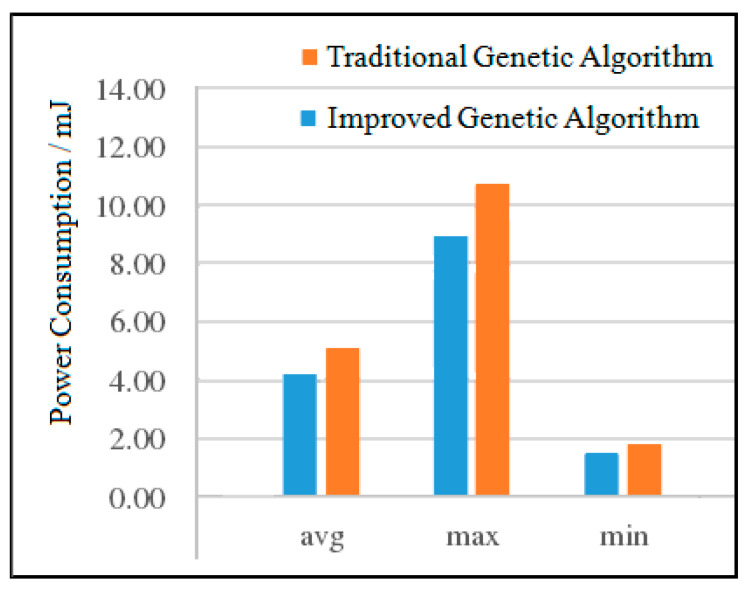
Comparison of the power consumption of the two algorithms for VOPD.

**Figure 15 micromachines-12-01217-f015:**
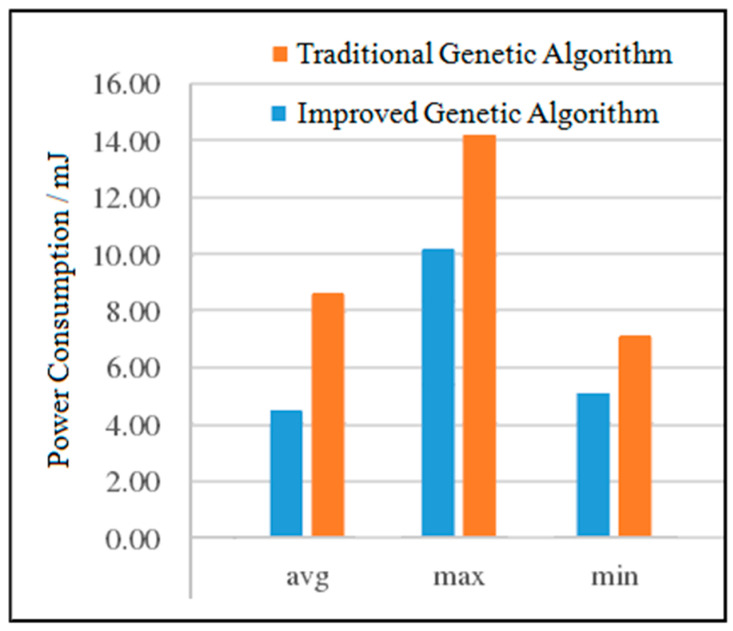
Comparison of the power consumption of the two algorithms for DVOPD.

**Figure 16 micromachines-12-01217-f016:**
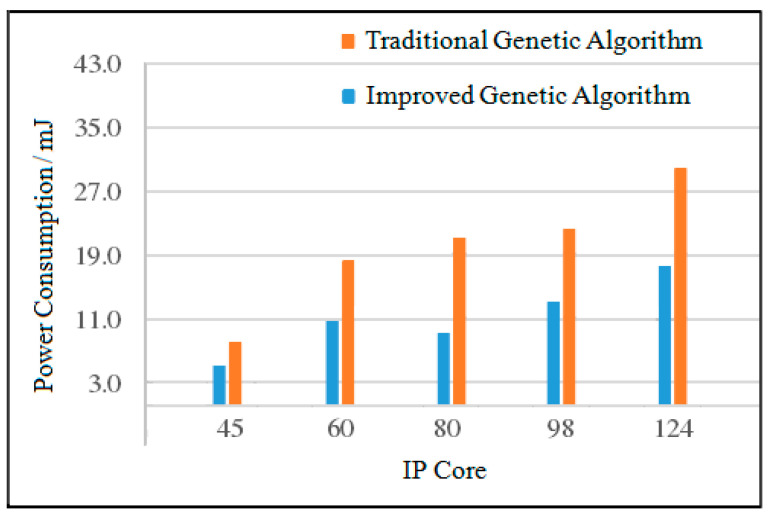
Comparison of average power consumption of the two algorithms.

**Figure 17 micromachines-12-01217-f017:**
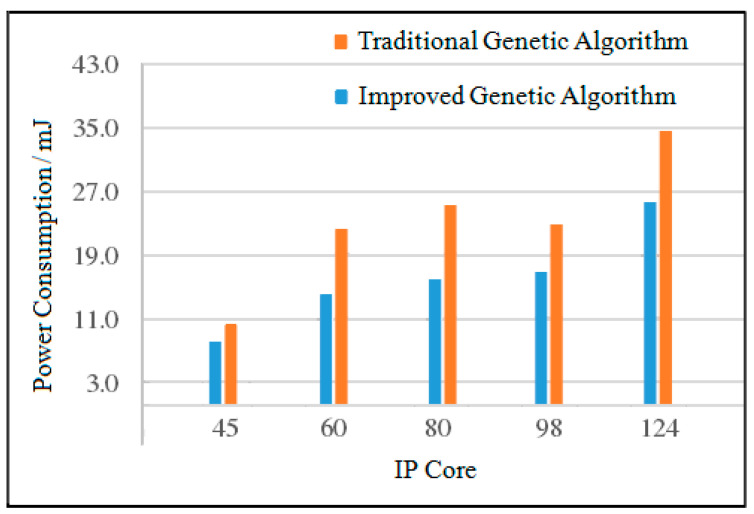
Comparison of the maximum power consumption of the two algorithm.

**Figure 18 micromachines-12-01217-f018:**
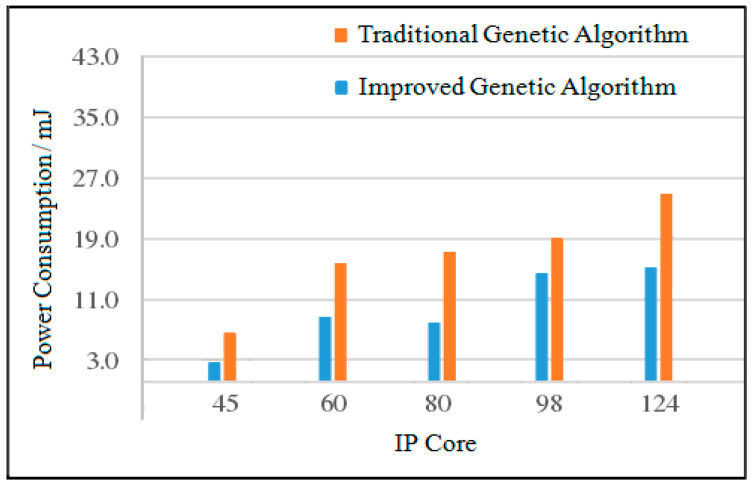
Comparison of the minimum power consumption of the two algorithms.

**Table 1 micromachines-12-01217-t001:** Random task graph power consumption comparison (mJ).

Algorithm	45 Cores			60 Cores		
	avg	max	min	avg	max	min
Traditional Genetic Algorithm	6517	9980	4968	18,040	23,644	16,870
Improved Genetic Algorithm	4120	8355	2800	11,000	14,730	8811

**Table 2 micromachines-12-01217-t002:** Random task graph power consumption comparison (Continued) (mJ).

Algorithm	80 Cores			98 Cores			124 Cores		
	avg	max	min	avg	max	min	avg	max	min
Traditional Genetic Algorithm	22,743	25,833	17,678	23,588	23,722	19,000	31,014	34,936	25,234
Improved Genetic Algorithm	9254	16,885	8677	14,320	17,455	15,000	17,911	26,002	15,005

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
