# Peer review of "3D NoC Low-Power Mapping Optimization Based on Improved Genetic Algorithm"

_micromachines, 2021, doi:10.3390/mi12101217_

Round 1

Reviewer 1 Report

Actually, researches on 3D NoC Mapping are adequate, your introduction about the topic is tedious and similar to some other papers. Also, using the annealing algorithm in this area has been explored by other researchers. And comparing with the traditional way is not convincing, I think some SOTA solutions should be your challenger and comparer. Therefore, the paper lacks the innovation to be published.

Author Response

We would like to thank you for your careful reading and comments.In most cases ,benchmark and baseline should be the most basic comparison objects of an paper. The motivation of a paper comes from wanting to surpass the existing baseline and benchmark. The experimental data of the paper should be based on baseline and benchmark to judge whether there is improvement. The only difference is that baseline pays attention to a set of methods, while benchmark prefers the highest indicators at present, such as precision, recall and other quantifiable indicators. For example, in the NLP task, BERT is the current SOTA. If we have an idea that can surpass BERT, the baseline to be compared in the experimental part of the paper is BERT, and the benchmark to be compared is the specific indicators of BERT. In our paper, we have verified the energy saving and feasibility of the algorithm through experiments, so we think the algorithm proposed in this paper may be still have the practical significance of publication.

Reviewer 2 Report

The authors present an alternative to traditional genetic algorithms, which is used to study the mapping and mainly energetic performance of NoC 3D.

The results show an outstanding performance in terms of energy consumption using the modified genetic algorithm, which is desirable.

The work is outstanding and provides a significantly useful tool in the design of NoCs.

The only recommendation is the following: The word thesis is used repeatedly, which I am not sure if it should be changed to paper or work.

Author Response

We thank the reviewer for reading our paper carefully and giving the above positive comments.According to the reviewer's opinion, we replaced the word thesis in the text with work or paper according to the actual context.We hope our revised manuscript can be accepted for publication.

Reviewer 3 Report

Compared to the first and second versions of the article, which I had previously reviewed, this work has become much better. Nevertheless, there are still a number of drawbacks in it:

  1. Formatting error of paragraph alignment (lines 90-95).
  2. The formula is part of a sentence and must end with a dot (lines 115, 118, 122, 148).
  3. Letters in formulas should be italicized (compare lines 111 and 122 - there should be consistency everywhere).
  4. How is the maximum number of iterations chosen?
  5. Figure 13 needs to be revised, and an adequate vertical axis should be made.
  6. Is there section 1 after section 4.2.3 on line 393? How can subsections and sections be distinguished?
  7. Line 487. After a colon, enumeration in the form of a list is implied, but there is not any list.
  8. In references, not all initials have dots (lines 539, 541, 543, 545, etc.). And sometimes, even spaces are missing (line 551 “Pande PP.”).

Author Response

We would like to thank you for your careful reading, helpful comments, and constructive suggestions, which has significantly improved the presentation of our manuscript.
We have carefully considered all comments and revised our manuscript accordingly. We added the missing points and spaces mentioned in the comments,and corrected all the clerical errors including font and format problems.
Here are our responses to some other comments.

How is the maximum number of iterations chosen?
Response:About selecting the maximum number of iterations, the operation efficiency of the algorithm, the number of samples of the parent, the number and quality of the children need to be considered. If the number of iterations is too small, the population is not mature and the algorithm is not easy to converge; If the number of iterations is too large, the population is too early, the algorithm is already skilled, and it is easy to increase the time overhead and waste resources. Therefore, in this experiment, the number of genetic iterations is set to 200.

Figure 13 needs to be revised, and an adequate vertical axis should be made.
Response:In order to visually compare the power consumption of the three structure diagrams, the measurement unit (scale) of the three diagrams is set to the same, which is milliJoule. If there is any modification, the scale must be modified. The diagram may be more significant, but it is not easy to compare, which is easy to misunderstand the readers.

Is there section 1 after section 4.2.3 on line 393? How can subsections and sections be distinguished?
Response:Our paper is written in strict accordance with the template of the journal, sections and subsections are also distinguished according to the template of the journal.

We believe that our responses have well addressed all concerns from the reviewers. We hope our revised manuscript can be accepted for publication.

Round 2

Reviewer 1 Report

Please check the attached pdf, which is published in the Journal of Computer-Aided Design & Computer Graphics. My judgment was based on that I don't think there's significant innovation between yours and hers. Please illustrate this. After checking your institution, I think you can read this Chinese paper.
